# Scalable Intersectional Bias Auditing in Vision-Language Models through Combinatorial Interaction Testing

**Heejin Bin**[*], **Junyoung Choi**[*], **JangHyun Kim**[*], **Seungjae Kim**[*], **Shin Yoo**
School of Computing, KAIST
Daejeon, South Korea
{heejbin, vsdpqksvs3, big01ad, kimseung, shin.yoo}@kaist.ac.kr

## Abstract

Intersectionality analysis is critical for algorithmic fairness, since individuals hold multiple and overlapping identities, leading to unique challenges and biases. However, scalable detection of intersectional fairness bugs, i.e., systematic misbehaviors that emerge only in higher-order intersectional subgroups, remains difficult due to the scarcity of data across diverse identity combinations. We propose a scalable auditing framework for intersectional biases in Vision-Language Models (VLMs): our framework is based on Combinatorial Interaction Testing (CIT) and diffusion models. CIT enables systematic sampling of all $t$-way identity interactions with a minimal set of test suites, while diffusion allows us to generate specific inputs for VLMs that fit the given combination of identities. By integrating CIT with synthetic image generation, we substantially mitigate the computational and generative burden, making the exploration of deeply nested subgroups tractable and scalable. Our empirical evaluation shows that our approach can flexibly balance subgroup specificity with test efficiency, uncovering compounding biases that remain invisible to conventional univariate or bivariate assessments.

## 1 Introduction

Current fairness research acknowledges that bias does not arise from a single axis of inequality alone; rather, it emerges from complex, layered, and interlocking systems of oppression. As established in the framework of intersectionality(Crenshaw, 1989), social identities like race and gender are not independent variables but are intrinsically linked within a broader social hierarchy; thus, analyzing these attributes in isolation fails to capture the intertwined power relations that define them. For instance, Hamidieh et al. (2024) have demonstrated that Vision-Language Models (VLMs) often exhibit biases where the term 'Homemaker' is not only disproportionately associated with women over men, but also overwhelmingly linked to Indian women compared to other racial groups. Furthermore, Shukla et al. (2025) suggest that efforts to de-bias a model for a single axis, such as gender, can inadvertently amplify biases along another, such as race, highlighting the critical need for a holistic, intersectional perspective.

Despite the consensus on its importance, evaluating intersectional fairness presents a critical challenge rooted in the tension between categorical specificity and statistical reliability. As researchers subdivide identity categories, they encounter a combinatorial explosion that results in a prohibitively large number of potential intersections. This increased resolution inevitably makes direct data collection for each group nearly impossible. Further, as noted by several previous studies (Himmelreich et al., 2025; Wang et al., 2022; Islam et al., 2023; Molina & Loiseau, 2022), the more specific a subgroup becomes, the smaller its sample size becomes. Such extreme sparsity reduces these intersectional groups to isolated data points, which in turn lose their representative power. Consequently, these underrepresented identities remain as systemic blind spots in fairness evaluation.

Due to these challenges, existing fairness benchmarks often limit their scope to single axis — such as gender, race, or age — or combinations of two or three attributes. To overcome this problem,

---

[*]Equal contribution

we propose a systematic and affordable framework for exploring high-dimensional intersectional biases in VLMs. Our approach resolves the 'data-as-points' issue using diffusion-based synthetic generation, and manages the combinatorial explosion by adopting *Combinatorial Interaction Testing (CIT)* (Kuhn et al., 2004), enabling efficient navigation of a large intersectional space.

Our framework redefines intersectional auditing as a systematic search for "fairness bugs" of VLMs within the high-dimensional search space of visual identities. We introduce a three-stage analysis strategy. First, we apply Combinatorial Interaction Testing to Diffusion prompts, collapsing millions of potential intersections into a tractable set of thousands of representative test cases. This ensures broad coverage of the interaction space without the overhead of brute-force testing. Second, by leveraging these CIT-optimized prompts, we generate synthetic images that encapsulate multiple identity intersections simultaneously, allowing for an efficient and affordable evaluation. Finally, our framework analyzes the extent to which each intersectional subgroup identity is recognized and accurately represented by the VLMs. By measuring how effectively the model identifies diverse identity axes within a visual context, we can algorithmically detect where the model fails to acknowledge specific intersections, uncovering deep-seated biases that simpler benchmarks overlook. [1]

## 2 BACKGROUND

**Intersectionality**, a term coined by Crenshaw (1989), captures a concept long articulated by earlier scholars of color to describe how identities such as race, gender, and class overlap to create unique modes of discrimination and privilege. There have been numerous instances where technologies interact with pre-existing societal inequities, resulting in outcomes that disproportionately burden historically marginalized communities(Buolamwini & Gebru, 2018; Obermeyer et al., 2019; Rodriguez, 2023; Rodà et al., 2025). In response, researchers have begun to adopt intersectionality as a theoretically-grounded methodology to mitigate outcome bias and inspire machine learning research.(Islam et al., 2023; Kearns et al., 2018)

**Combinatorial Interaction Testing**, or Combinatorial Testing was first proposed by Kuhn et al. (2004). Empirical analysis of fault data across diverse domains demonstrates that most software failures are triggered by specific combinations of only one or two system input parameters. Based on these empirical results, if all faults in a system are triggered by a combination of $n$ or fewer inputs, then testing all $n$-way combinations becomes effectively equivalent to exhaustive testing. A single test case contains a massive number of $t$-way combinations. For example, a test case consisting of 20 input parameters simultaneously covers $\binom{20}{4} = 4845$ different 4-way combinations. By constructing a minimal *Covering Array* that encompasses all possible n-way interactions, we can achieve comprehensive fault detection with a manageable and mathematically optimized number of test cases, achieving pseudo-exhaustive search.

## 3 PREVIOUS WORK

### 3.1 VISION-LANGUAGE MODEL FAIRNESS

Extensive research has focused on the fairness of Vision-Language Models (VLMs), revealing significant biases regarding identities such as gender, skin tone, age, and ethnicity (Wu et al., 2024). Addressing these internal biases is not merely a localized improvement but a fundamental necessity for the integrity of the broader AI ecosystem, as VLMs increasingly function in two critical, yet high-risk, capacities.

First, as evaluative frameworks, VLMs are frequently utilized to audit other models when ground-truth labels are cost-prohibitive. For instance, Zhang & Chunara (2024) used CLIP to analyze associations between images and sensitive attributes, while Shukla et al. (2025) leveraged VLMs for automated labeling to evaluate Text-to-Image models. If the VLMs themselves are biased, it risks masking or distorting the unfairness of the target models, leading to false assurances of safety.

Second, as foundational components, VLMs are integrated into diverse downstream applications. In this role, their internal biases are not static but actively transferred to derivative systems, embed-

---

[1]Code can be found at `https://github.com/heejbin/Combinatorial-VLM-Bias`

Table 1: Identity axes and attribute values used for intersectional prompt generation

| Axes(Abbr.) | Attribute Type | Labels |
|---|---|---|
| Race&Ethnicity(ETH) Faith Socioeconomic Status Age(AGE) Gender(GEN) Disability Status(DIS) | Protected | European, African, East-Asian, South-Asian, Latino Christian, Muslim, Hindu, Buddhist, Jew, Unreligious Upper Class, Middle Class, Lower Class Infant, Child, Teenager, Adult, Middle-aged, Elderly Male, Female Without a disability, With a visual impairment, With autism, With cerebral palsy, With a hearing impairment, With a prosthetic limb, Who uses a wheelchair |
| Appearance(APP) Location(LOC) | Contextual | Fat, Thin, Attractive, Unsightly, Strong, Weak Urban area, Suburban area, Rural area, Remote area, Neighborhood, City, Workplace, School |

ding systemic prejudices into the very architecture of the ecosystem. Consequently, ensuring the internal fairness of VLMs is essential to prevent cascading bias where flawed evaluations and biased foundations reinforce one another.

## 3.2 INTERSECTIONALITY IN FAIRNESS STUDIES

Recent fairness research emphasizes intersectionality, noting that bias often manifests at the intersection of multiple attributes where marginal fairness fails to capture granular disparities (Filippi et al., 2023). This perspective has been applied across diverse domains, from clinical language models to recommendation systems (Ramachandranpillai et al., 2024; Wang et al., 2024; Suk & Han, 2025; Wastvedt et al., 2024). However, practical auditing faces a combinatorial explosion; as the number of protected attributes $n$ increases, the subgroup search space grows exponentially, leading to extreme data sparsity. Consequently, prior evaluations have been largely confined to a few primary axes like race, gender, and age, leaving sparse intersectional subgroups as "long-tail outliers" that are difficult to collect or annotate.

To address these challenges, prior strategies have focused on leveraging shared statistical patterns (Wang et al., 2022), robust fairness estimation via batch-based learning (Islam et al., 2023), or establishing statistical bounds between marginal and intersectional fairness (Molina & Loiseau, 2022). While these offer valuable insights, they primarily optimize the learning process or rely on predefined domain assumptions regarding attribute correlations. Such approaches are often insufficient for the high-dimensional landscape of Vision-Language Models (VLMs), where the fundamental hurdle is not just the learning process itself, but the complete absence of balanced, fine-grained intersectional data. To fill this gap, we shift the focus to a generative auditing framework. By combining diffusion-based synthesis to actively populate the sparse intersectional space with CIT to manage complexity, we provide a scalable solution that operates independently of prior data distributions or manually defined assumptions.

## 4 METHODOLOGY

### 4.1 SYNTHETIC IMAGE GENERATION WITH CIT

#### 4.1.1 IDENTITY AXES

The selection of axes aims to balance a comprehensive coverage of diverse social identities with the practical requirements of large-scale image generation. As summarized in Table 1, these dimensions are strategically categorized based on their roles in fairness literature and their impact on intersectional analysis:

- **Protected Attributes**: These include core identity markers such as *Race&Ethnicity*, *Age*, *Gender*, *Disability Status*, *Faith*, and *Socioeconomic Status*. While aligned with established

fairness research, these attributes are specifically expanded to encompass dimensions critical to intersectionality.

- **Contextual Attributes**: These comprise situational and visual factors such as *Appearance* and *Location*. Although not always classified as inherent identities, they fundamentally shape an individual's lived experience and how they navigate the world. We incorporate these to analyze how such contexts manifest in model representations and interact with protected identities for a more ecologically valid analysis.

For a more detailed discussion on the selection axes and the theoretical grounding for each axis, please refer to Appendix A.1.

### 4.1.2 PROMPT TEMPLATE DESIGN

To encode these multi-axial identities into natural language, we adopt a structured prompt template designed for semantic clarity and consistency: *"a photo of a {Appearance} {Race&Ethnicity} {Age&Gender} in a {Location}, who is of {Faith} faith, has a {Socioeconomic status} status, and {Disability status}."*

This template is structured to ensure linguistic naturalness and grammatical coherence across all attribute combinations. For instance, we jointly encode *Age* and *Gender* and carefully order each attribute to avoid clinical phrasing and effectively represent the intended intersectional identities.

### 4.1.3 COMBINATORIAL TESTING FOR PROMPT REDUCTION AND IMAGE GENERATION

The full Cartesian product of all selected axes results in an intractably large number of images and queries to VLMs. To efficiently explore the vast space of identity combinations, we employ *Combinatorial Interaction Testing (CIT)*. By leveraging CIT's ability to detect failures triggered by sparse factor interactions, we systematically sample representative intersectional cases while maintaining manageable test suite sizes.

To navigate the high-dimensional space of identity combinations, we apply $t$-way combinatorial testing. This approach reduces the exhaustive search space of over 18 million possible queries into computationally feasible, representative subsets while preserving intersectional coverage up to order $t$. We generate $t$-way covering arrays for $t \in \{2, 3, 4\}$, resulting in 3,450 (69), 23,450 (469), and 142,350 (2,847) images (prompts), respectively. By constructing these minimal covering arrays, we achieve a pseudo-exhaustive evaluation of complex identity interactions at a fraction of the original computational cost. All images are generated via Stable Diffusion; for detailed technical configurations, please refer to Appendix A.2.2.

### 4.2 PRELIMINARY VALIDATION: DIFFUSION MODEL RELIABILITY

To ensure the generated images can serve as a reliable ground truth for evaluating VLMs, we conduct a preliminary qualitative review of the generated images. 590 images — 10 images per prompt from the 2-way sets — were manually inspected by authors to check whether the visual representations consistently reflect the intended prompt attributes.

This process leads to several refinements based on the generative capabilities of the model:

- **Attribute Exclusion**: *Faith* and *Socioeconomic Status* have been excluded from the final analysis. While these dimensions may possess visual components, they are not sufficiently or consistently manifested in the generated outputs to serve as reliable ground truths.
- **Scope Refinement**: For *Disability Status* and *Appearance*, we narrow the scope to "Who uses a wheelchair/Without a disability" and "Fat/Thin", respectively, as the generative model fails to consistently produce recognizable representations for other sub-categories. Following this refinement, we maintained high fidelity for Disability Status (94.00%, up from 69.31%) and Appearance (91.96%, significantly improved from the initial 43.66% alignment).

The refined image set achieves 91.96% overall alignment rate. Specifically, high fidelity has been maintained for *Gender* (99.42%), *Age* (94.64%), and *Disability Status* (94.00%). These results

suggest that prompt attributes can reasonably serve as ground truth labels, indicating that VLM failures are likely driven by internal biases despite some remaining ambiguity.

## 4.3 VLM EVALUATION PROTOCOL

To audit the perceptual fairness of VLMs, we specifically evaluate the pre-trained Vision-and-Language Transformer (**ViLT**) (Kim et al., 2021) architecture. We measure how consistently the model recognizes a specific identity marker across different intersectional combinations. The goal is to determine if the presence of certain attributes interferes with or alters the recognition of others.

Based on the preliminary validation, we formalize the evaluation space using the $K = 6$ refined identity axes that demonstrated high visual reliability. We define the set of these validated axes as $\mathcal{A} = \{A_1, A_2, \ldots, A_6\}$, where each axis $A_k$ represents a distinct identity domain (e.g., $A_{GEN}$, $A_{ETH}$). For each input image $I_i$, we denote the ground truth labels from the generation prompt as a 6-dimensional attribute vector:

$$\mathbf{a}_i = (a_{i,1}, a_{i,2}, \ldots, a_{i,6}) \tag{1}$$

where $a_{i,k} \in A_k$ is the ground truth attribute value of the person in $I_i$ for the $k$-th axis. This vector serves as the "pseudo-ground truth" for auditing VLM recognition performance across diverse intersectional contexts.

To isolate the model's perceptual performance from linguistic confounding factors, we generate standardized queries using a fixed template. For each attribute value $a_{i,k}$ in the vector $\mathbf{a}_i$, we construct a query $Q(a_{i,k})$: *"Is the person $\{a_{i,k}\}$?"*. During the inference phase, the VQA model generates un-normalized probability scores $p_{\text{yes}}$ and $p_{\text{no}}$ for the binary response tokens "Yes" and "No", respectively. To ensure a focused evaluation of these two choices, we compute a normalized affirmative probability $\tilde{p}_{i,k}$ as follows:

$$\tilde{p}_{i,k}(I_i, a_{i,k}) = \frac{p_{\text{yes}}}{p_{\text{yes}} + p_{\text{no}}} \tag{2}$$

The discrete decision $\hat{y}_{i,k}$ is obtained by applying a classification threshold:

$$\hat{y}_{i,k} = \mathbb{1}(\tilde{p}_{i,k} \geq 0.5) \tag{3}$$

A perfectly fair and robust model is expected to yield $\hat{y}_{i,k} = 1$ with high confidence ($\tilde{p}_{i,k} \approx 1$) across all attribute axes, provided the image content is clear. The probability $\tilde{p}_{i,k}$ serves as a continuous metric for the model's perceptual certainty. By statistically analyzing these scores across the diverse demographic groups defined within the $K = 6$ axes, we identify "fairness bugs".

# 5 RESULTS

## 5.1 EVALUATION METHOD

Our evaluation framework adapts the concept of *Explicit Bias* — originally formulated as a generative failure in Text-to-Image models (Luo et al., 2024) — to the domain of VLM recognition. We investigate the extent to which a VLM accurately perceives and validates each demographic attribute when it is part of a multi-axial intersectional identity. For example, we examine whether the recognition of the attribute "Female" remains consistent when intersected with different age groups (e.g., "Elderly Female(elderly woman)" vs. "Child Female(girl)", or how the presence of a specific *Disability* status affects the model's ability to acknowledge other protected attributes. This approach allows us to uncover systematic fairness gaps where the model fails to maintain a robust representation of specific identity dimensions under diverse intersectional conditions, effectively leading to the perceptual erasure of marginalized sub-populations.

## 5.2 FAIRNESS METRICS FOR COUNTERFACTUAL ATTRIBUTE PROBING

These metrics are designed to detect biases by monitoring how model predictions fluctuate when demographic contexts change. The goal is to ensure that the model's understanding and representation of a target attribute is grounded in visual features rather than being sensitive to intersecting demographic variations.

Table 2: Baseline attribute recognition accuracy performance by axes

| Category | Baseline Accuracy | Category | Baseline Accuracy |
|---|---|---|---|
| Race&Ethnicity | 0.7848 | Age | 0.7575 |
| Appearance | 0.2162 | Gender | 0.8193 |
| Location | 0.8433 | Disability Status | 0.5178 |

### 5.2.1 ATTRIBUTE RECOGNITION ACCURACY (ACC)

As a baseline, we report the standard attribute recognition accuracy for each attribute axis $A_k$. Using the discrete prediction $\hat{y}_{i,k}$ and the pseudo-ground truth framework, the accuracy is defined as:

$$\text{ACC}(A_k) = \mathbb{E}_{i \sim \mathcal{D}} \left[ \mathbb{1}(\hat{y}_{i,k} = 1) \right] \tag{4}$$

where $\mathcal{D}$ denotes the generated images. This metric measures the model's ability to correctly identify the presence of the ground-truth attribute $a_{i,k}$ in image $I_i$. By monitoring $\text{ACC}(A_k)$, we ensure that fairness improvements are not artifacts of uniformly degraded model performance across the axes.

### 5.2.2 INTERSECTIONAL COUNTERFACTUAL SENSITIVITY

To quantify how much the model's confidence in a *Target Axis* ($A_{target}$) depends on a *Varying Axis* ($A_{vary}$), we define the *Intersectional Counterfactual Sensitivity*. For a given set of *Fixed Axes* ($S_{fixed}$), the flip rate is the average absolute difference in the mean affirmative probability $\mu$ of the target attribute across all pairs of values in the varying axis:

$$\text{Flip}(A_{target} \mid A_{vary}, S_{fixed}) = \mathbb{E}_{S_{fixed}} \left[ \mathbb{E}_{g_m, g_n \sim A_{vary}} |\mu_{m,S} - \mu_{n,S}| \right] \tag{5}$$

where $\mu_{m,S} = \mathbb{E}[\tilde{p}(I, a_{target}) \mid A_{vary} = g_m, S_{fixed}]$ is the mean affirmative probability for the target attribute $a_{target} \in A_{target}$, given a specific value $g_m$ from the varying axis and a fixed context $S_{fixed}$. A high sensitivity indicates that the model's certainty is heavily dependent on demographic variations rather than invariant visual features, signaling a violation of statistical parity.

### 5.2.3 INTERSECTIONAL COUNTERFACTUAL DISPARITY

The *Intersectional Counterfactual Disparity* measures the maximum performance disparity in recognizing the *Target Axis* caused by the *Varying Axis*. Within a matched context of *Fixed Axes*, it is defined as:

$$\text{Gap}(A_{target} \mid A_{vary}, S_{fixed}) = \mathbb{E}_{S_{fixed}} \left[ \max_{g_m \in A_{vary}} \alpha_{m,S} - \min_{g_n \in A_{vary}} \alpha_{n,S} \right] \tag{6}$$

where $\alpha_{m,S} = \mathbb{E}[\hat{y} \mid A_{vary} = g_m, S_{fixed}]$ represents the target attribute recognition accuracy for the target axis when the varying axis is set to $g_m$ under context $S_{fixed}$.

This metric isolates the bias by ensuring that the accuracy comparison is performed only between groups that share identical attributes across all $S_{fixed}$, thereby identifying systematic recognition failures tied to the interaction between $A_{target}$ and $A_{vary}$.

### 5.3 QUANTITATIVE RESULTS

Table 2 reports the baseline attribute recognition accuracy of the VLM model across individual demographic and contextual axes. Overall, the model achieves relatively high accuracy on attributes such as *Location*, *Gender*, and *Race&Ethnicity*, indicating that these attributes are, in isolation, largely recognizable from the generated images. In contrast, substantially lower performance is observed for *Appearance* and *Disability Status*, suggesting that these attributes pose greater perceptual challenges even under controlled synthetic settings.

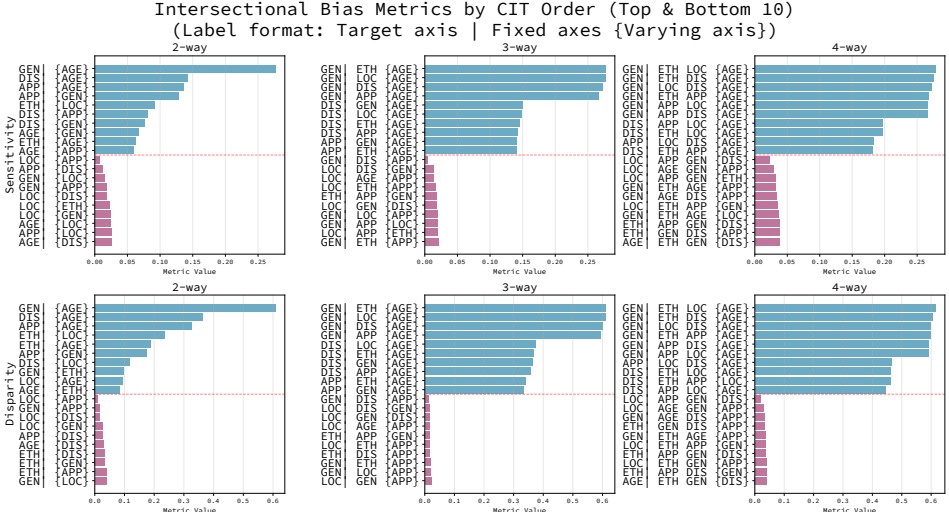

Figure 1: Intersectional counterfactual sensitivity and disparity across Top/bottom intersectional subgroups. Higher-order intersections expose severe disparities that are not observable under univariate evaluations. Subgroups above the red dotted line represent the top 10 subgroups (blue), while those below represent the bottom 10 subgroups (red) for each metric.

However, these aggregated accuracies provide only a coarse view of model behavior. While several protected attributes appear to be recognized reliably under the univariate evaluation, this perspective masks substantial heterogeneity that emerges when multiple attributes are considered jointly.

Figure 1 provides a summary of intersectional bias patterns revealed by the proposed metrics, reporting the top and bottom 10 intersectional groups for the *Intersectional Counterfactual Sensitivity* and *Intersectional Counterfactual Disparity* across 2-, 3-, and 4-way settings. Across all intersectional orders, a consistent trend emerges in which groups that vary *Age* target *Gender* exhibit the largest bias scores. This pattern becomes more pronounced when additional fixed attributes such as *Race&Ethnicity*, *Location* or *Disability* are included.This implies that the model may have been predominantly trained on specific age-gender archetypes, causing it to fail in representing or recognizing individuals who fall outside these overrepresented intersectional categories. As the order of intersectionality increases from two to four, the magnitude of both metrics grows, indicating that bias effects are compound under higher-order attribute interactions.

In contrast, while the absolute accuracy for *Appearance* and *Disability* remains relatively low, these groups appear among the lowest-scoring cohorts. This implies that the model's behavior, although suboptimal in accuracy, exhibits a degree of stability under these specific contextual perturbations, as opposed to the volatile accuracy shifts observed in other demographic intersections.

Overall, these results demonstrate that severe biases are concentrated in a small subset of intersectional configurations that would remain obscured under univariate evaluations. This analysis highlights the effectiveness of intersection-aware metrics in exposing biases that are not apparent from baseline accuracy alone, motivating the need for targeted, intersection-aware bias auditing.

## 5.4 QUALITATIVE ANALYSIS

As shown in Figure 2, initial univariate accuracy for *Age* indicates a significant deficiency within the "Child" demographic. However, a bivariate (2-way) analysis by *Gender* reveals that this performance gap is non-uniform; the suboptimal results are primarily driven by the "Female Child" subgroup, whereas the model maintains relatively robust accuracy for "Male Child".

To further investigate the nature of these disparities, a 3-way intersectional analysis is conducted, incorporating *Disability Status* (categorized as "who uses a wheelchair" vs. "without a disability"). The results demonstrate a compounded bias: while the error rate is already elevated for "Female child without a disability", the performance is most severely degraded for "Female child who uses

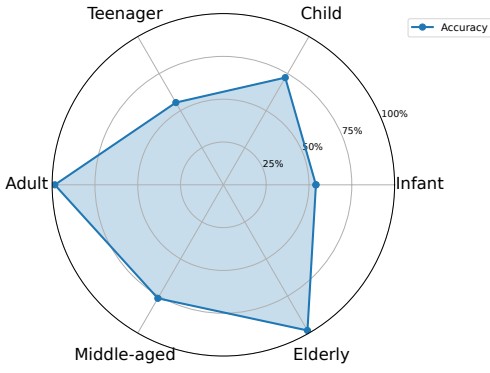

(a) Univariate analysis: Age accuracy across all demographics.

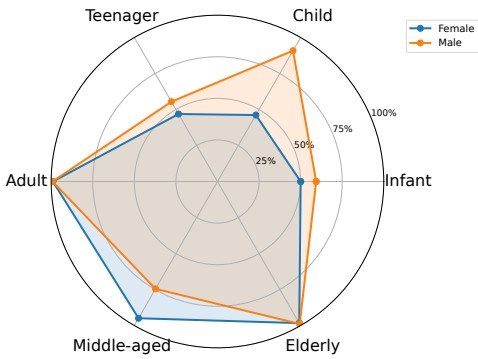

(b) Bivariate analysis: Age accuracy by gender.

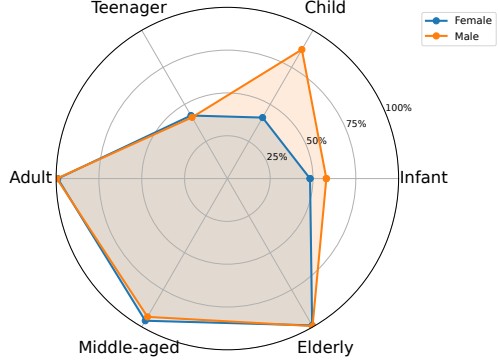

(c) Trivariate analysis: Age accuracy by gender for people who use wheelchairs.

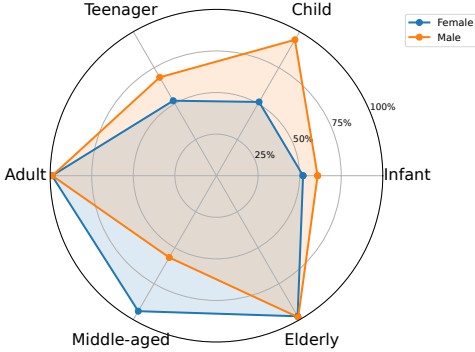

(d) Trivariate analysis: Age accuracy by gender for people without disability.

Figure 2: Multi-level intersectional analysis of Age estimation accuracy. (a) highlights the initial "Child" accuracy deficit; (b) identifies the "Female Child" subgroup as the primary driver of this trend; (c) and (d) reveal exacerbated performance degradation at the intersection of "Female" gender and "wheelchair" usage.

a wheelchair". These findings suggest that the intersection of *Age*, *Gender*, and *Disability* markers significantly exacerbates model bias, leading to the highest degree of inaccuracy for this specific subgroup. Visual examples of these failure cases, specifically illustrating the compounded bias for the 'Female child who uses a wheelchair' subgroup, are provided in Appendix A.2.2.

## 6 DISCUSSION AND LIMITATIONS

The observed biases in VLMs are not isolated errors but systematic deficiencies. The drops in recognizing and representing specific intersections reveal significant blind spots within the model. This indicates that high aggregated accuracy often masks severe performance degradation for diverse populations, with potential risks in real-world deployments.

Our framework further advocates for a shift from uniform data expansion toward a principled data prioritization strategy by utilizing synthetic data as a diagnostic tool. By leveraging CIT, practitioners can systematically identify high-risk and structurally vulnerable subgroups where the model remains unreliable, providing a scalable roadmap for data-centric interventions. Integrating this prioritization logic into the development lifecycle — through targeted data collection, sample synthesis, and fine-tuning — can ensure a more efficient allocation of resources to improve VLM robustness across diverse demographic intersections.

Despite the insights provided by our framework, this study has several limitations that open avenues for future research. First, the set of attributes used in our combinatorial auditing is not exhaustive. Factors such as *Religion* or *Socioeconomic Status*, though present in the prompts, were not explicitly analyzed, potentially introducing confounding effects. Additionally, the underlying diffusion models are not bias-free; we observed technical constraints such as structural warping in specific markers (e.g., wheelchairs). Due to computational resource constraints, isolating these variables through a complete re-generation of the dataset remains a task for future work.

While we implemented a rigorous human validation process, the shared cultural background of the authors may introduce subtle biases. Furthermore, our current framework discretizes *Gender* into a binary classification to ensure visual grounding. While we aimed to avoid imposing androgynous stereotypes, we acknowledge that excluding non-binary identities is a significant limitation. Future research should explore more nuanced alignment strategies that represent the fluid spectrum of gender identity without falling into reductive visual archetypes.

Regarding generalizability, while our auditing focused on the ViLT architecture, evaluating our findings across diverse VLM backbones, such as CLIP-based variants, is an essential next step. Additionally, while our human validation rigorously examined 2-way interactions, future work should scale this verification to higher-order combinations (e.g., 3-way or 4-way) to ensure robust visual grounding as combinatorial complexity increases.

Furthermore, as noted by Ovalle et al. (2023), much of the current literature—including this work—tends to treat intersectionality primarily as a task of optimizing fairness metrics for demographic subgroups. While our study identifies performance disparities, a more profound exploration of how specific attribute representations within VLMs lead to amplified bias in downstream tasks remains a challenge for future work. Beyond technical analysis, it is imperative to address the distributive harm these biases inflict on marginalized groups in real-world deployments. Consequently, future research should investigate targeted interventions for these "fairness bugs" to ensure more robust mitigation.

Finally, defining which attributes should be visually manifested remains an open representational question. While traits like *Age* are relatively visually apparent, others such as *Sexual orientation* may not (and perhaps should not) be explicitly rendered by both generative models and VLMs. Future studies should explore the spectrum of perceivability across different attributes in synthetic auditing to establish more ethical and effective bias-probing standards in vision models.

## 7 CONCLUSION

In this work, we present a scalable generative auditing framework to uncover intersectional biases in VLMs. By integrating **Combinatorial Interaction Testing (CIT)** with **Diffusion-based image synthesis**, we successfully navigate the combinatorial explosion of identity axes, transforming an intractable search space of several million potential intersections into a representative set of thousands of test suites. Our evaluation reveals that aggregate fairness metrics often mask severe "fairness bugs" that emerge only at higher-order intersections. We demonstrate that models maintaining high univariate accuracy can fail significantly when encountering compounding identities—such as the intersection of *Gender*, *Age*, and *Disability Status*—resulting in localized disparities (e.g., misidentifying a "female child who is using a wheelchair") that traditional reporting misses. Consequently, we advocate for a shift from coarse-grained auditing to a principled, intersection-aware framework. Despite challenges in representing non-perceivable attributes or generative biases, this approach provides a diagnostic roadmap for targeted data acquisition and fine-tuning, ensuring foundation models serve all populations equitably.

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

## A  APPENDIX

### A.1  RATIONALE FOR IDENTITY AXES AND ATTRIBUTE SELECTION

The selection of identity axes and their corresponding attribute labels follows a systematic process designed to bridge established fairness evaluation standards with the nuanced requirements of intersectional analysis. The taxonomy presented in Table 1 is developed through the following three stages:

1. **Consolidation of Existing Benchmarks**: We first identify core identity dimensions—such as *Race*, *Gender*, and *Age*—that are consistently utilized in previous fairness benchmarks(Luo et al., 2024; Hamidieh et al., 2024; Shukla et al., 2025). This ensures our results remain comparable with prior literature on algorithmic bias.

2. **Expansion via Global Frameworks**: To transcend the limitations of traditional univariate metrics, we incorporate additional axes such as *Disability Status*, *Faith*, *Location*, and *Socioeconomic Status*. These dimensions are selected based on the UN Women (2022), which emphasizes that systemic marginalization often occurs at the intersection of these frequently overlooked attributes. The specific attribute categories utilized for each identity axis are detailed in Table 1.

3. **LLM-Assisted Refinement and Visual Filtering**: For each axis, we define labels that are both semantically distinct and visually manifestable. We utilize an LLM to brainstorm an extensive pool of potential attributes. This pool is then manually curated to exclude traits that lack consistent visual signifiers or might force the model to rely on reductive stereotypes. For instance, while we recognize gender as a spectrum, *non-binary* identities are excluded from the image-based evaluation. In this study, gender is operationalized as a binary category (Male/Female) to ensure a stable visual ground truth for auditing. While this allows for systematic combinatorial testing, we acknowledge that this discretization does not capture the full spectrum of gender identities. A detailed discussion on the implications of this scope and future directions for including non-binary identities is provided in Section 7

By discretizing each axis into these curated labels, we enable a systematic, combinatorial approach to prompt generation. This allows for the evaluation of model performance across both common and underrepresented identity intersections, ensuring a more ecologically valid assessment of generative bias.

## A.2 Experimental Reproducibility and Technical Details

### A.2.1 Combinatorial Test Suite Generation

To ensure the reproducibility of our sampling methodology, we will provide the source code used to generate the $t$-way covering arrays. The combinatorial test suites are generated using Randomized Greedy Covering implemented in our framework.

For each interaction strength $t \in \{2, 3, 4\}$, the covering arrays are constructed to ensure 100% coverage of all $t$-way combinations of identity axes. The resulting test cases are then transformed into natural language prompts using a predefined template.

### A.2.2 Image Generation Configuration

Synthetic images are produced via the **Stable Diffusion XL (SDXL 1.0-base)** model using the HuggingFace *diffusers* framework.

**Hyperparameters** The generation process utilizes 15 inference steps with a classifier-free guidance (CFG) scale of 7.0. All images are generated at a resolution of $1024 \times 1024$ pixels. To account for the stochastic nature of the diffusion process and to ensure statistical robustness, we generate 50 independent images per single prompt.

**Prompt Engineering** A standardized positive suffix is appended to all CIT-generated prompts to maintain consistent composition across multiple batches:

> ", exposed face, looking at the camera, ultra quality, sharp focus, 8k, photorealistic, intricate details, hdr, high resolution"

To mitigate common artifacts such as disfigured limbs or excessive cropping that might obscure identity-related features (e.g., mobility aids or ethnic attire), we apply the following negative prompt (Ban et al., 2024):

> "lowres, bad anatomy, error body, error hands, missing fingers, cropped, worst quality, low quality, jpeg artifacts"

**Hardware** All generation tasks are performed on a single 20GB partition of an NVIDIA A100 GPU (GRID A100D-20C). VQA evaluation is performed on a NVIDIA RTX 5080 GPU.

## A.3 FAILURE CASES

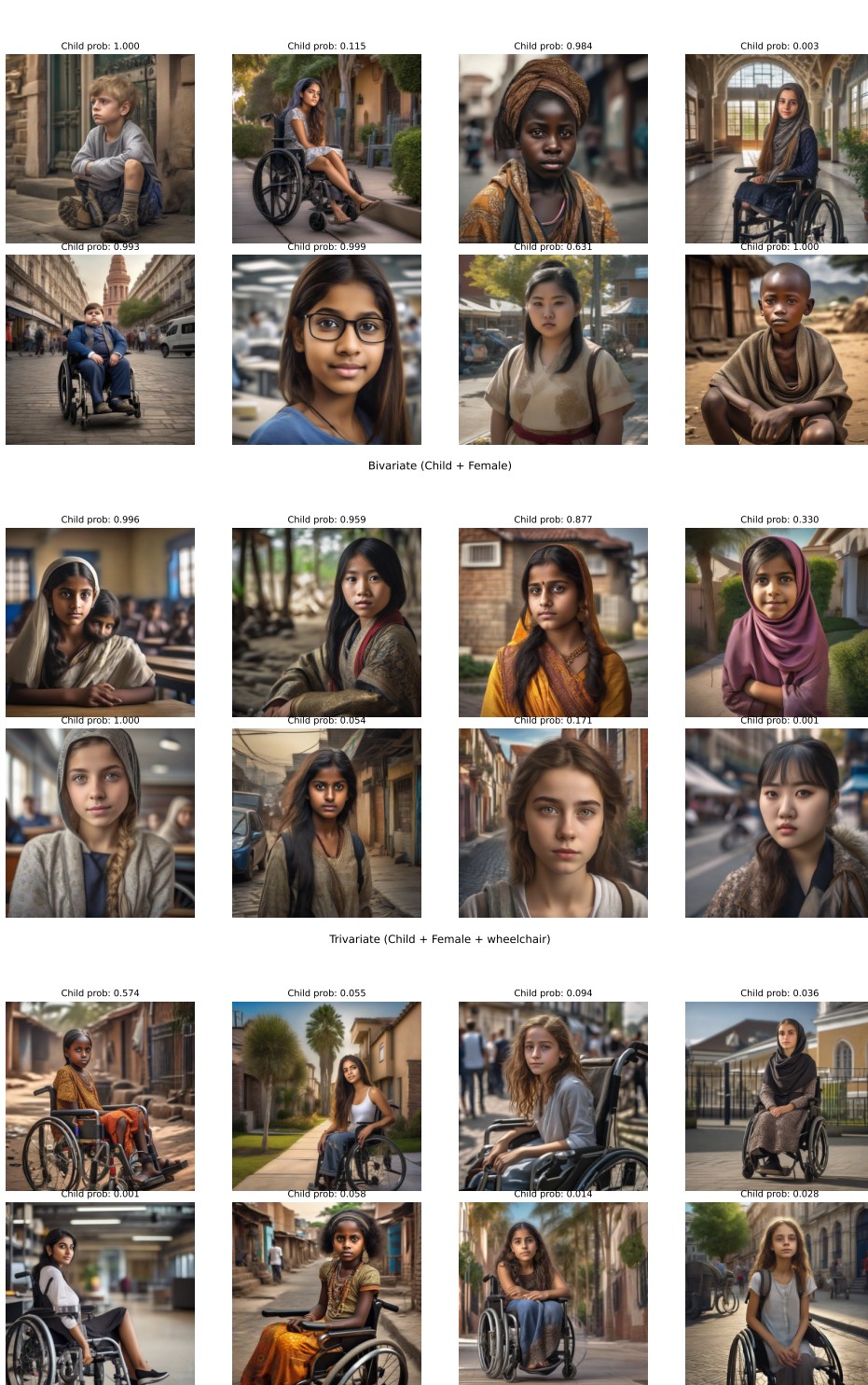

Figure 3: Qualitative analysis of VLM recognition confidence across increasing intersectional complexity. The model's predicted probability for the 'Child' category progressively declines.

