# OpenReview forum: "Scalable Intersectional Bias Auditing in Vision-Language Models through Combinatorial Interaction Testing"
_ICLR.cc/2026/Workshop/AFAA — AFAA 2026 Poster_

### Official Review · Reviewer_9LNJ · 2026-02-12
**Much needed framework - should have more evidence included in appendix**

**Rating:** 4
**Confidence:** 4

**Summary:**

The paper introduces and discusses a scalable auditing framework for intersectional biases in Vision Language Models. Their framework is based on combinatorial interaction testing (CTI) and diffusion models.

**Strengths:**

The paper understand the importance of intersectionality in auditing AI fairness. It identifies that biases can be studied more deeply at the intersection of multiple axes of identity and marginality. Based on this understanding, the paper introduces fairness metrics to evaluate performance (and bias) of vision language models in generating intersectional identities.

**Weaknesses:**

The paper gives detailed evidence for quantitative analysis, but does not go in-depth of qualitative analysis. More qualitative evidence will support their argument of mis or inept representing of intersectional marginalized public. Also, the paper does not give any example figures to showcase the evidence of their claims. The paper would benefit from an appendix of generated images with evidence of intersectional failures.

---

### Official Review · Reviewer_ZgNy · 2026-02-17
**Efficient Intersectional Bias Auditing in VLMs using Combinatorial Interaction Testing and Synthetic Images**

**Rating:** 4
**Confidence:** 5

**Summary:**

The paper is about testing vision language models to uncover fairness failures that only show up when multiple identity attributes overlap (e.g. age + gender + disability). The authors mentioned that intersectional auditing is hard due to following reasons: (1) Real data for rare identity combinations is not easily available and hence data sparsity is an issue. (2) Number of possible combinations explodes into millions which could lead to higher testing costs.
The authors propose a two stage process: Combinatorial Interaction Testing (CIT) to select a compact set of identity combinations (2/3/4 way interactions), and diffusion-based image generation (SDXL) to synthesize images matching those combinations. The authors found that ViLT with standardized yes/no questions like “Is the person {attribute}?” and quantify failures using accuracy plus two intersection aware metrics (counterfactual sensitivity and counterfactual gap). The main claim is that high order intersections (3-way or 4-way) expose “fairness bugs” that are invisible under standard univariate or bivariate fairness checks.

**Strengths:**

1. Applying a proven fault finding paradigm i.e. covering arrays (CIT) to fairness auditing to reduce a space of 18 million queries to a smaller set of 2847 prompts for 4 way coverage.
2. Use of diffusion to address the long tail of rare intersections data during evaluation.
3. Figure 2 provides a breakdown (univariate to trivariate) to clearly show how aggregate data masks specific subgroup failures.

**Weaknesses:**

1. There should have been multiple strong baselines than a single transformer based ViLt to validate finding generalize. They could have preferred latest visual models like CLIP variansts.
2. The manual inspection covers 590 images from 2-way only out of 142,350 images. There should have been a validation across 3-way and 4-way sets to confirm whether fidelity remain high at higher interactions.

---

### Official Review · Reviewer_YwjL · 2026-02-23
**Pressing work uncovering intersectionality biases in VLMs that is a good fit for AFAA 2026.**

**Rating:** 4
**Confidence:** 4

**Summary:**

The paper presents a thorough analysis on observed biases in Visual Language Models (VLMs) in pre-trained Vision-and
Language Transformer (ViLT) that causes systemic deficiencies across an intersection of diverse identity categories.

It utilizes Combinatorial Interaction Testing (CIT) with Diffusion-based image synthesis Stable Diffusion XL (SDXL1.0-base) to provides a scalable generative auditing framework to uncover intersectional biases in VLMs and  in turn results in severe performance degradation for diverse populations, with potential risks in downstream tasks and real-world applications.

The potential use-cases right from targeted synthetic data collection, sampling of fine-tuning models to handle underrepresented groups and ensure equitable distribution is also listed out to scale across fairness and intersection-aware systems.

Overall, this work perfectly aligns with the AFAA 2026 workshop goals of cultural and contextual dimensions, ethical and accountability frameworks as well as bias mitigation in foundation models to name a few and hence is a good accept.

**Strengths:**

1. The paper is very structured, there is a good balance and logical flow across sections and the background literature work on intersectionality and motivation for fairness is also emphasized well.

2. By using Combinatorial Interaction Testing (CIT), the search space is reduced to more representative and tractable test cases across multiple identity axes which can help in initial identification and evaluation without n exponential blow-up.

3. Preliminary validation does Attribute Exclusion and Scope Refinement across identity categories to further improve the model reliability.

4. Evaluation methodology proposes 3 metrics in Attribute Recognition Accuracy (ACC), Intersectional Counterfactual Sensitivity (Flip) and Intersectional Counterfactual Disparity (Gap). specifically designed to maintain a robust representation of specific identity dimensions under diverse intersectional conditions which uncovers systematic fairness gaps in case of model failure.

4. The quantitative results and analysis shows how bias effects are compounded under higher-order attribute interactions.

5. The limitations both from technical standpoint as well as the author's own inherent biases are also mentioned along with future directions of research for the study and its larger ethical implications.

6. Lastly, the authors promise to share the code upon acceptance which is good from a reproducibility and transparency standpoint.

**Weaknesses:**

A. Major Concerns
1. The significance of the red dotted line (to segregate Appearance and Location attributes from the rest) as well as the use of 2 different colour bars for the Top / Bottom Intersectional Subgroups in Figure1 is not clear so mention that explicitly.

2. How happens when the categories are ordered differently in the prompt template. Like, did any previous linguistic or fairness work to arrive the proposed coherent structuring?

3. While the rationale to exclude ”non-binary” gender identities as not having a singular or universal visual representation is highlighted in the Appendix and how it can possibly lead to androgenous stereotypes is postulated but it is not backed by any actual findings. Moreover, by leaving out such an important category in today's time is very limiting of the evaluation work. At least some possible measure to include this should be proposed in future directions.

B. Minor Improvements
1. Add relevant citation(s) as applicable in line 73 to 77 to highlight 'numerous instances' and ' In response, researchers have begun to adopt', especially given that you also have the last line which has space to add more words.

2. In line 206, High fidelity is reported across Gender, Age and Disability Status after refinements but what was the alignment rate before for these categories to understand the percentage change / delta improvement?

3. What are the values in each category prompts like say for example, Age - Elder, Younger or more precise 50 year old, etc.? It is the only attribute which becomes clear in Figure 2. So considering including this in the Appendix for other identity categories as well.

4 Missing text / sentence incomplete in lines 314-315.

---

### Meta-Review · Area_Chair_QfW1 · 2026-02-26

**Recommendation:** Main Papers Track
**Confidence:** 5

**Metareview:**

The paper proposes a scalable auditing framework for intersectional bias in vision-language models (evaluated on ViLT) by combining Combinatorial Interaction Testing (CIT) to select a compact set of identity-attribute combinations with diffusion-based image synthesis (SDXL) to generate test cases, and introduces intersection-aware metrics (accuracy plus counterfactual sensitivity/flip and counterfactual disparity/gap). Reviewers appreciate the clear structure and motivation, the use of CIT to avoid combinatorial explosion, and quantitative results indicating that bias effects can compound under higher-order interactions. The main concerns are about validation breadth and interpretability. Overall, the reviews are positive and I recommend acceptance.

---

### Decision · Program_Chairs · 2026-03-02

Accept (Poster)